# Effectiveness of Repetitive Transcranial Magnetic Stimulation (rTMS) Add-On Therapy to a Standard Treatment in Individuals with Multiple Sclerosis and Concomitant Symptoms of Depression—Results from a Randomized Clinical Trial and Pilot Study

**DOI:** 10.3390/jcm12072525

**Published:** 2023-03-27

**Authors:** Mohammad Ahmadpanah, Shiva Amini, Mehrdokht Mazdeh, Mohammad Haghighi, Alireza Soltanian, Leila Jahangard, Amir Keshavarzi, Serge Brand

**Affiliations:** 1Department of Clinical Psychology, School of Medicine, Hamadan University of Medical Sciences, Hamadan 6517838636, Iran; 2Behavioral Disorders and Substance Abuse Research Center, Hamadan University of Medical Sciences, Hamadan 6517838636, Iran; 3Department of Neurology, Faculty of Medicine, Hamadan University of Medical Sciences, Hamadan 6517838636, Iran; 4Department of Biostatistics, School of Public Health, Modeling of Non-Communicable Diseases Research Center, Hamadan University of Medical Sciences, Hamadan 6517838636, Iran; 5Center for Affective, Stress and Sleep Disorders (ZASS), Psychiatric University Hospital Basel, 4002 Basel, Switzerland; 6Division of Sport and Psychosocial Health, Department of Sport, Exercise and Health, University of Basel, 4052 Basel, Switzerland; 7Sleep Disorders Research Center, Health Institute, Kermanshah University of Medical Sciences, Kermanshah 6714869914, Iran; 8Substance Abuse Prevention Research Center, Kermanshah University of Medical Sciences, Kermanshah 6714869914, Iran; 9School of Medicine, Tehran University of Medical Sciences (TUMS), Tehran 1419733141, Iran; 10Center for Disaster Psychiatry and Disaster Psychology, Psychiatric Clinics of the University of Basel, 4002 Basel, Switzerland

**Keywords:** rTMS, sham condition, depression, multiple sclerosis, fatigue, EDSS

## Abstract

Background: Compared to the general population, persons with multiple sclerosis (MS) are at increased risk of suffering from major depressive disorder (MDD). Repetitive Transcranial Magnetic Stimulation (rTMS) was used successfully to treat individuals with MDD. Here, we conducted a randomized clinical trial and pilot study, and tested the effectiveness of rTMS adjuvant to a standard pharmacological treatment among persons with MS, compared to a sham condition. Materials and Methods: A total of 40 persons with MS (mean age: 32 years; 42.5% females; median EDSS score: 4) and with moderate to severe symptoms of depression were randomly assigned to the rTMS or to the rTMS sham condition, always as adjuvant intervention to the standard treatment with sertraline, a selective serotonin reuptake inhibitor (SSRI). rTMS consisted of 10 sessions each of 37.5 min; the sham condition was identical to the active condition except for the absence of rTMS stimuli. At the beginning and two weeks after the end of the study, participants reported on their fatigue, while experts rated the severity of participants’ depressive symptoms (Montgomery–Asberg Depression Rating Scale; MADRS), cognitive performance (Montreal Cognitive Assessment; MoCA), and degree of disability (Expanded Disability Status Scale; EDSS). Results: Data were analyzed per intent-to-treat. Scores for depression, fatigue, and EDSS declined significantly over time (large effect sizes), but more so in the rTMS condition than in the sham condition (large effect sizes for the time by group-interactions). Compared to the sham condition, scores for depression were significantly lower in the rTMS condition. Scores for cognition improved over time in both study conditions (large effect size). Conclusion: Compared to a sham condition, adjuvant rTMS to a standard pharmacological treatment ameliorated typical MS-related symptoms (depression; fatigue; EDSS scores). Results from this pilot study suggested that rTMS might be routinely applied in persons with MS displaying symptoms of depression and fatigue.

## 1. Introduction

Multiple sclerosis (MS) is a chronic and progressive disease of the central nervous system, and the third leading cause of neurological disabilities worldwide [1]. MS affects an estimated five to three hundred people per hundred thousand worldwide [2]. This chronic neurological disease most commonly affects adults between 20 and 50 years of age, with an average age of onset in the early fourth decade of life [3]. Furthermore, a higher preponderance of MS was consistently reported for females [4]. In MS, the immune system shows an inflammatory reaction against nerve tissue and causes tissue demyelination [5]. Although the pathogenesis of the disease was not fully elucidated, it is believed that the clinical manifestations are due to an immune response that crosses the blood–brain barrier (BBB) and enters the central nervous system [6]. As a consequence, MS and associated CNS damage result in impairments of various aspects of functioning.

Not surprisingly, studies showed that persons with MS experience much higher levels of mental health problems, such as symptoms of depression, stress, and anxiety, than the general population [7]. More specifically, two systematic reviews and meta-analyses revealed a higher prevalence of symptoms of major depressive disorder (MDD) in individuals with MS than in the general population [8,9]. There are three possible reasons for these higher prevalence rates. First, symptoms of depression may occur in response to immune and inflammatory changes [10]. Second, atrophy and brain lesions may lead to mood disorders [10,11,12], in particular when the temporal lobe is affected [13]. Third, depression and anxiety can also be responses to the chronic illness and its psychosocial, financial, and job-related consequences. Thus, compared to the general population, individuals with MS have higher rates of job loss and unemployment [14,15,16]. These are in turn related to a reduced quality of life in the general population, but also among individuals with major depressive disorder [17], and among individuals with MS, more specifically [18].

The following interventions were assessed as treatments for the symptoms of depression in individuals with MS [19]: nine psychological and psychotherapeutic interventions, and three interventions with MDD-specific medications reduced depressive symptoms, when compared to control or placebo conditions. More specifically, psychotherapeutical interventions such as cognitive behavioral therapy (CBT) and mindfulness-based stress reduction (MBSR) reduced symptoms of depression (and anxiety), when compared to controls conditions, while selective serotonin-reuptake inhibitors (SSRI) were administrated as psychopharmacological treatments [19].

Surprisingly, however, it appears that the beneficial use of repetitive Transcranial Magnetic Stimulation (rTMS) in the treatment of symptoms of depression in individuals with MS was not considered. The aim of the present pilot study was, therefore, to investigate the impact of a standardized rTMS intervention on symptoms of depression, compared to a sham condition.

Repetitive Transcranial Magnetic Stimulation (rTMS) is a safe and non-invasive method in which magnetic pulses affect the cortical activity of the stimulated area [20]. This stimulation is repeated and performed in very short time units. High frequency magnetic pulses (above 5 Hz) stimulate neuronal activity, while low frequency magnetic pulses (less than 1 Hz) reduce this activity. In addition, rTMS impacts on changes of both glucose levels and neurotransmitter activity in individuals with MDD [21].

Recent studies showed that high-frequency rTMS on the dorsolateral prefrontal cortex alters the activity of glutamatergic neurons [22]. Changes in glutamate concentrations are not limited to the stimulated region. There are also increases in the right dorsolateral prefrontal cortex and in the left cingulate cortex [21].

Next, rTMS is a well-established and standardized treatment for individuals with MDD [23]. However, as regards MS, previous studies focused on the possible beneficial effects of non-invasive brain stimulation techniques such as tDCS (transcranial direct-current stimulation), rTMS, and ECT (electroconvulsive therapy) on MS-related symptoms such as mobility and gait [24,25,26]. One study examined TMS as a treatment of depressive symptoms for neurological disorders in general, but not for MS specifically [27]. In their review, Palm et al. [24] concluded that, for individuals with MS, ECT was effective in the treatment of severe psychiatric disorders, while tDCS and ECT targeted at depressive symptoms, fatigue, tactile sensory deficits, pain, motor performance, and spasticity had mixed results. Somaa, de Graaf, and Sack [26] proposed in their statement paper that TMS has the power to impact positively on a broad range of symptoms related to neurological diseases, including MS. Iodice, Manganelli, and Dubbioso [25] reviewed 21 studies on the effect of tDCS and rTMS on symptoms of MS and concluded that such interventions could lead to improvements with respect to fatigue, motor performance, spasticity, and cognitive performance, while results were mixed for pain, sensory deficits, and bladder function. Iodice, Manganelli, and Dubbioso [25] concluded that both tDCS and rTMS were promising non-invasive brain stimulation techniques still understudied in individuals with MS. Kan et al. [28], in their systematic review and meta-analysis, examined twenty-five randomized controlled trials on the efficacy of non-invasive brain stimulation techniques among individuals with MS; of those, six were rTMS studies. These studies showed a reduction in muscle spasticity, while other effects were observed for MS-associated pain and symptoms of impaired mood.

Overall, the literature reviewed above suggests that rTMS might have the power to ameliorate MS-related symptoms, including symptoms of depression, though findings were mixed and inconclusive. Considering this, the aims of the present pilot study were to evaluate the effectiveness of adjuvant rTMS to the standard treatment for individuals with MS, compared to an rTMS sham condition. Following Godfrey, Muthukumaraswamy, Stinear, and Hoeh [23], Fregni and Pascual-Leone [27], and Somaa, de Graaf, and Sack [26], we expected that symptoms of depression and fatigue, along with severity of disability and cognitive functions, would show positive change with rTMS, compared to the sham condition. In testing this possibility, we believe that the present study could have the power to shed light on the efficacy of rTMS adjuvant to the standard MS-related medication treatment for typical MS-related health issues, always compared to a sham condition.

## 2. Method

### 2.1. Participants

Individuals diagnosed with MS and self-reporting symptoms of depression were approached to participate in the present intervention study. Participants were all informed about the aims of the study and the confidential data handling. Thereafter, participants signed a written informed consent.

Inclusion criteria were: 1. aged 18 years or older; 2. diagnosis of MS, as ascertained by an experienced neurologist not otherwise involved in the study, and based on the McDonald criteria [2,29,30]; 3. Expanded Disability Status Scale (EDSS) score of <7 [31]; 4. mild to moderate major depressive disorder, as ascertained by an experienced psychiatrist not otherwise involved in the study, and based on a thorough clinical interview [32]; 5. standard and individualized treatment of symptoms of MS; 6. signed written informed consent. Exclusion criteria were: 1. severe psychiatric disorders such as bipolar disorders, schizophrenia spectrum disorder, substance use disorder, or neurodegenerative disorders, again as ascertained by an experienced psychiatrist, and based on a thorough clinical interview [32]; 2. undergoing any kind of brain stimulation therapy; 3. history of epilepsy or brain tumors; 4. rTMS contraindication such as shunts or any other metal object near the head that could not be removed from the patient; 5. pregnant or breastfeeding women.

### 2.2. Procedure

This clinical trial and pilot study was performed between 2021 and 2022 at the Farshchian Hospital in Hamadan (Iran). The Ethics Committee of Hamadan University of Medical Sciences and the Iranian Registry of Clinical Trials (IRCT), with the IR.UMSHA.REC.1400.101 and IRCT20160523028008N11 IDs, approved the study.

After the thorough clinical examination, and once inclusion and exclusion were established, participants were randomly assigned either to the rTMS or to the sham condition. To this end, 20 red and 20 blue cards were each sealed in separate envelopes; envelopes were placed in an opaque ballot box and stirred. An assistant drew an envelope, assigned the participant to the indicated study condition, and the envelope was then destroyed. The interventions and sham conditions are described below.

### 2.3. Sample Size Calculations

To calculate the sample size, we relied on effect sizes reported in Kim et al. [33], and we used the G*Power software [34]. The expected f effect size was 0.33, alpha was 0.05, power (1-beta) = 0.95; number of groups: 2; number of measurements: 2; the calculated total sample was 32, though to cover for possible drop-outs, 20 participants were recruited for each group.

### 2.4. Measures

#### 2.4.1. Sociodemographic and MS-Related Information

At the beginning of the intervention, participants reported their age (years), gender at birth (male; female), disease duration (years), civil status (single; married), and highest educational level completed (compulsory school diploma; high school diploma; higher educational degree).

#### 2.4.2. Fatigue

At the beginning and at the end of the intervention, participants rated their degree of fatigue on a 10-point Likert scale, with 1 indicating no fatigue and 10 indicating a very intense fatigue [35].

#### 2.4.3. Depression Severity

At the beginning and at the end of the study, and to rate a participants’ depression severity, experts (psychiatrists; clinical psychologists) not otherwise involved in the study used the Farsi version [36] of the Montgomery–Asberg Depression Rating Scale (MADRS) [37]. Items refer to typical symptoms of depression such as visible and reported sadness, impaired sleep, lack of energy, and suicidal ideation. Answers are given on 7-point rating scales ranging from 0 (=no signs at all) to 6 (=almost always; persistently), with higher sum scores reflecting more severe depression.

#### 2.4.4. Cognitive Performance

At the beginning and at the end of the study, experts rated participants’ performance with the Farsi version [38] of the Montreal Cognitive Assessment (MoCA) [39]. The MoCA is widely used to screen for mild cognitive deficits. It consists of 30 items, focusing for instance on working/short-term memory, long-term memory, and executive functions, and is completed in about 30 min. A higher score reflects better cognitive performance.

#### 2.4.5. Degree of Disability

Experienced neurologists blind to the study condition to which participants were assigned to rated their degree of MS-related impairment. To this end, they used the Expanded Disability Status Scale (EDSS) [40]. The EDSS is an accepted and widely used tool for expert-rated and objective assessment of disability levels of individuals with MS. The total score is on a scale from 0 to 10, with increments of 0.5–1.0, and with higher scores reflecting a higher level of disability. EDSS scores between 1.0 and 4.5 refer to people who are fully ambulatory. EDSS scores between 5.0 and 9.5 reflect impairment to ambulation. Meyer-Moock et al. [31] reported in their systematic review the high validity and reliability of the EDSS, and concluded that the EDSS is suitable for describing clinical status and degree of physical disability, and for monitoring disease progression.

## 3. Interventions

### 3.1. Medications

Patients were individually treated with MS-related medications. In addition, patients received sertraline tablets for 4 weeks at a dose of 25 mg to 100 mg. The initial dose of sertraline was 25 mg daily; if tolerated, 25 mg was added to the initial dose each week. The maximum prescribed dose during one month of treatment was 100 mg and the average prescribed dose was 50 mg.

### 3.2. Repetitive Transcranial Magnetic Stimulation

For rTMS, as in previous studies [41,42,43], a 70 mm Double Air Film Coil Magstim (Magstim Company Ltd., Spring Gardens, Withland, Carmartenshire, UK) was employed.

The coil-to-skull distance was kept to the minimum possible.

As described elsewhere [44,45], the following standard protocol was applied:

A high frequency stimulation was conducted at 10 Hz, with pulse duration of 4 s, 26 s intertrain interval (resulting in an overall duration of 37.5 min [46]), in 3000 total pulse to the left dorsolateral prefrontal cortex (DPFC) with stimulation intensity of 110% of the resting motor threshold (RMT). The RMT was calculated by visible muscle movements (twitch) in the collateral muscle of the right thumb (abductor pollicis brevis), as reported by the International Federation of Clinical Neurophysiology (IFCN) [47,48].

The prefrontal cortex localization was targeted by the “5 cm rule”, that is to say: the left DLPFC was localized 5 cm anterior to cortical motor areas on a line parallel to the parasagittal line.

Collectively, each participant underwent the intervention or sham condition for ten sessions; each session lasted for 37.5 min. The frequency of ten sessions was determined as follows: Feffer et al. [49] showed that, among individuals with MDD, early symptom improvements after ten sessions predicted the overall outcome of rTMS interventions. Similarly, an early response to rTMS was associated with clinical improvements among individuals with MDD [50], while an early non-response to rTMS was associated with an overall non-response to rTMS [51]. Höppner et al. [22] observed an antidepressant efficacy of rTMS interventions after 10 out of 12 days. Kazemi et al. [52] employed bilateral rTMS among individuals with MDD and observed clinically relevant improvements after 10 sessions. Brunelin et al. [53] concluded from their literature review that short courses of five to ten sessions of rTMS reduced symptoms of depression among individuals with MDD to a clinically significant degree. Considering this background, the decision was to employ ten sessions of rTMS to reduce symptoms of depression among individuals with MS.

rTMS sessions occurred between 8 am and noon under the supervision of a psychiatrist not otherwise involved in the study. While a TMS technician determined the default setting (active rTMS vs. sham) of the rTMS device, the patient was unaware of whether she/he underwent active rTMS or sham rTMS. While, out of necessity, the technician responsible for the interventions was not blind to the intervention condition, the technician was neither involved in randomization nor in the psychological assessment, data acquisition, or statistical data elaboration.

For the sham condition, the stimulation was applied at the same site of active treatment but with only the side edge resting on the scalp. Furthermore, as for the treatment group, stimulation was applied at high frequency (10 Hz) with a duration of 37.5 min. Next, sham TMs generated sound clicks and bodily sensations virtually identical to the active treatment condition, though with no effects on neuronal processes. Additionally, while the coil orientation in the rTMS condition was 45° relative to the interhemispheric line, the coil orientation in the sham condition was angled differently and placed at a 90° angle away from the skull in a tilted position. Neither participants in the active condition nor participants in the sham condition had visual feedback of the coil orientation and coil position; considering this, all participants were blind to the intervention condition.

After every session, participants were asked to remain seated for a further 15–20 min to avoid possible rTMS-related side effects such as vertigo.

### 3.3. Statistical Analysis

To compare MS-related and sociodemographic indices between participants in the active and sham condition, two *t*-tests and three X^2^-tests were performed.

To compare symptoms of depression, fatigue, and cognitive performance between the two study conditions at baseline, three *t*-tests were performed.

To compare the EDSS scores between the two study conditions at baseline, a U-test after Mann–Whitney was performed.

To compare the dosage of sertraline between the two study conditions at study end, a *t*-test was performed.

To calculate changes in depression, fatigue, cognitive performance and degree of disability (EDSS score) over time and between and within the study conditions, a series of ANOVAs for repeated measures was performed. Factors were: study condition: active vs. sham intervention; time: begin and end of the study, and the study condition by time—interaction. Dependent variables were fatigue, depression severity, cognitive performance, and degree of disability. In addition to *p*-values, we calculated effect sizes. A *p*-value of <0.05 was considered as significant. The data analysis was performed by intent-to-treat (ITT) with the last observation carried forward (LOCF), and not per protocol. All statistical calculations were performed with SPSS^®^ 28.0 (IBM Corporation, Armonk, NY, USA) for Apple Mac^®^.

## 4. Results

### 4.1. General Information

Figure 1 is the flow-chart reflecting the number participants approached, the number of participants included and randomized in the study, and the number of participants completing the study. Briefly, from the 2 × 20 participants assigned to the two study conditions, 17 completed the study in the sham condition, and 18 completed the study in the active condition. As mentioned, the statistical analysis was performed per intent-to-treat (ITT) with the last observation carried forward (LOCF), and not per protocol.

Next, Table 1 reports the descriptive and statistical indices of MS-related and sociodemographic dimensions between participants in the active and participants in the sham condition. No descriptively or statistically significant differences were observed for age, illness duration, gender, civil status, or highest educational level completed. Furthermore, by the end of the intervention, study conditions differed neither descriptively not statistically as regards sertraline dosage.

### 4.2. Symptoms of Depression, Fatigue, Cognitive Performance, and Degree of Disability from the Beginning to the End of the Study and between Participants in the Active rTMS and rTMS Sham-Condition

At baseline, there were no descriptive (see Table 2) or statistically significant differences between the rTMS and sham conditions for symptoms of depression, fatigue, or cognitive performance (t-values between 0.34 and 1.33; *p*-values between 0.20 and 0.73). At baseline, EDSS scores did not differ descriptively (see Table 2) between the rTMS and the sham condition, nor were differences statistically significant (Mann–Whitney U: 226.50, *p* = 0.47).

As regards the dosages of sertraline at the end of the study, no descriptive or statistically significant differences were observed between the rTMS and the sham condition (rTMS: m = 80.00, SD = 19.19; sham: m = 78.75, SD = 20.32; t(38) = 0.20, *p* = 0.84).

Table 2 and Table 3 report the descriptive and inferential statistical indices for symptoms of depression and fatigue, and cognitive performance and degree of disability.

Over time, symptoms of depression and fatigue decreased (large effect sizes), but more so in the active than in the sham condition (large effect size for the interaction).

Over time, cognitive performance improved (large effect size), with no other improvements or decrease between or within study conditions.

Over time, degree of disability decreased (large effect size), but more so in the active than in the sham condition.

## 5. Discussion

The aims of the present pilot study were to determine whether repetitive Transcranial Magnet Stimulation (rTMS) could reduce symptoms of depression and fatigue and improve cognitive performance, and whether rTMS could decrease degree of disability, in each instance compared to a sham condition. Results confirmed these possibilities, though not for the cognitive performance, which changed over time, but irrespective from the study condition. The novelties of the present results add to the current literature in that rTMS appeared to have a positive impact on a broad variety of core symptoms that are characteristic of individuals with MS.

### 5.1. rTMS and Depression

In the present pilot study, neuronal and neurophysiological concentrations were not assessed. Considering this, we relied for an explanation of the positive impact of rTMS on symptoms of depression on previous studies of healthy controls and of individuals with depression resulting from other types impairments. Studies among healthy individuals indicated that high-frequency rTMS on the lateral dorsal cortex of the forehead altered the activity of glutamatergic neurons [54]. Importantly, the change in glutamate concentrations was not limited to the stimulation area; these also increased in the right lateral dorsal cortex of the right forehead and the left cingulate cortex. For individuals with MDD, recent studies showed that rTMS produced changes in neural glutamate concentrations [55]. In this view, changes in glutamate levels were also dependent on levels before stimulation: patients with MDD with low glutamatergic concentrations experienced a larger increase in concentrations after rTMS sessions than did MDD patients who already had higher glutamate concentrations at baseline [23].

In addition, numerous studies examined the effect of rTMS on individuals with symptoms of depression, though results were mixed. In one study of individuals with post-stroke depression, 35 participants received rTMS, and 35 received routine treatment. After 8 weeks of treatment, serum levels of IL-1β, IL-6, and TNF-α in the intervention group significantly reduced [56]. A decrease in inflammatory markers was understood to be an improvement in depression, linking symptoms of depression to inflammatory markers [57,58,59,60]. Kim et al. [33] evaluated the efficacy of rTMS in treating peripartum depression. Eleven pregnant women in the second or third trimester were included in the rTMS group, and 11 others were included in the control condition with no specific interventions. Participants in the intervention group received 20 sessions of rTMS. Over time, Hamilton Depression Rating Scale (HDRS) scores decreased significantly but more so in the rTMS condition than in the control condition, and the response rate in the intervention condition was 81.8%, while it was 45.4% in the control condition [33]. Similar effects were reported in other studies [23,57,61,62,63,64,65], though as far as we know, the present pilot study is the first successfully to apply rTMS to symptoms of depression among individuals with MS.

### 5.2. rTMS and Fatigue

Fatigue is one of the most common symptoms in individuals with MS, and fatigue significantly impacts on their quality of life. Results from the present pilot study showed that rTMS, compared to the sham condition, reduced symptoms of fatigue. Krogh et al. [63] reported in their meta-analysis that the impact of rTMS on fatigue among individuals with MS was mixed. In contrast, another meta-analysis [62] of 14 studies observed a consistent positive impact of rTMS on symptoms of fatigue among individuals with MS. The evidence from the present pilot study is a better fit with the findings of this latter meta-analysis [62].

### 5.3. rTMS and Cognitive Performance

Compared to the general population, individuals with MS display more cognitive problems. Cognitive impairment is possibly associated with a relative loss of brain volume [66]. In the present pilot study, we observed that cognitive function increased over time in both groups but found no advantage for the active over to the sham condition. The evidence available from the study is unable to illuminate the underling physiological and psychological mechanisms. It is possible that participants, irrespective of assigned condition, felt more secure and supported, which might, in turn, have had a positive effect on their cognitive performances. Findings from previous work showed that lower scores for anxiety predicted higher marks among university students [67]. Additionally, symptoms of depression were consistently associated with poorer cognitive processes [68,69]. It is, therefore, possible that the cognitive improvements observed among all participants were related to the general improvement in symptoms of depression. In this regard, we also note that, over time, there were reductions in symptoms of depression among all participants, but more so in the rTMS condition.

### 5.4. rTMS and Disability

To our knowledge, whether rTMS has the potential to improve EDSS scores as a proxy for degree of disability was not systematically investigated as a primary outcome variable among individuals with MS. Three meta-analyses among individuals with neurological disorders concluded that improvement could be expected [24,25,26], though the underlying mechanisms are so far unclear. Nonetheless, the present study demonstrated that rTMS can reduce degree of disability, when compared to a sham condition.

The novelty of the results should be balanced against the following limitations. First, the sample size was relatively small, though we did prioritize effect sizes which are not sensitive to sample sizes. Second, inclusion and exclusion criteria were such that the sample was highly homogeneous; this degree of homogeneity does not necessarily reflect everyday clinical experience. As such, it remains unclear whether the improvements observed in the present study would transfer to the wider population of individuals with MS with more heterogenous sociodemographic characteristics and MS-related symptoms. Third, there is good evidence that regular physical activity and exercising impacts positively on a broad variety of MS-related symptoms [70,71,72,73], including sleep patterns [70,71,72,73,74,75,76]. Consequently, future studies should consider both physical activity/exercising and subjective sleep as possible confounders.

## 6. Conclusions

Symptoms of depression, fatigue, and disability improved over time, but more so in the rTMS condition than in the sham condition. For cognitive performance, rTMS did not have further advantages. rTMS should, therefore, be considered more often when treating individuals with MS who have MS-specific health issues.

## Figures and Tables

**Figure 1 jcm-12-02525-f001:**
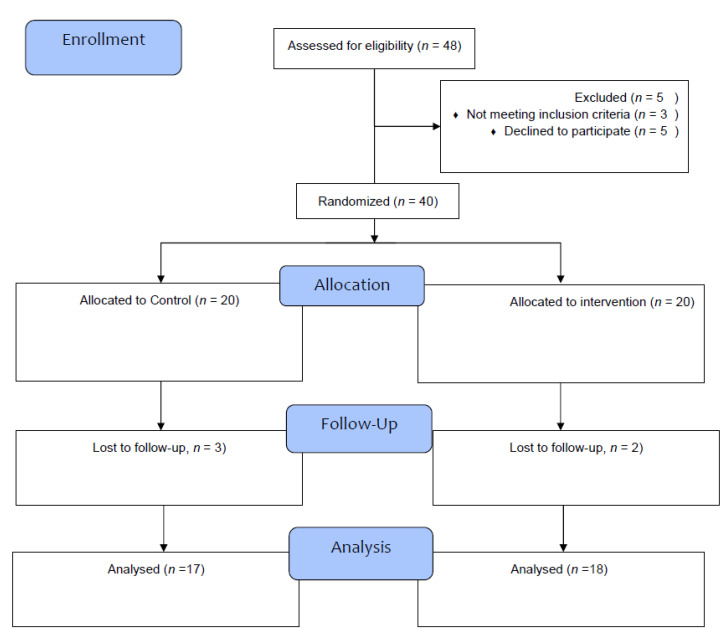
Flow-chart reflecting the number of participants at each state of the study.

**Table 1 jcm-12-02525-t001:** Sociodemographic information of participants in the rTMS and sham condition.

		Study Conditions		Statistics
Variable		rTMS	Sham	
N		20	20	
		M (SD)	M (SD)	
Age (years)		32.05 (6.82)	31.95 (6.93)	t(38) = 0.05
Duration disease (years)		8.88 (2.88)	7.67 (2.04)	t(38) = 1.52
		n (%)	n (%)	
Gender	Female	12 (30)	11 (27.5)	X^2^(N = 40, df = 1) = 0.10
	Male	8 (20)	9 (22.5)	
Marital status	Married	15 (37.5)	13 (32.5)	X^2^(N = 40, df = 1) = 0.48
	Single	5 (12.5)	7 (17.5)	
Education	Under diploma	6 (15)	3 (7.5)	X^2^(N = 40, df = 2) = 1.30
	Diploma	8 (20)	10 (25)	
	Academic degree	6 (15)	7 (17.5)	

**Table 2 jcm-12-02525-t002:** Descriptive statistical overview of symptoms of depression and fatigue, cognitive performance and the degree of disability at baseline and at the end of the study, separately for the rTMS and for the sham condition.

	Baseline		Study End	
	rTMS	Sham	I rTMS	Sham
N	20	20	20	20
	M (SD)	M (SD)	M (SD)	M (SD)
Depression	28.72 (2.72)	28.94 (2.97)	23.17 (2.92)	25.86 (4.06)
Fatigue	6.17 (0.75)	6.29 (0.59)	5.75 (0.97)	6.26 (0.62)
Cognitive performance	24.22 (1.80)	24.88 (1.58)	26.00 (1.78)	25.70 (1.80)
	Median (range)	Median (range)	Median (range)	Median (range)
Disability ^1^	4.75 (2.00–6.50)	3.50 (2.50–6.00)	3.50 (2.00–5.50)	3.50 (3.00–5.50)

Notes: ^1^ EDSS (Expanded Disability Status Scale) scores, an ANOVA was also computed, while scores are reported as medians and ranges.

**Table 3 jcm-12-02525-t003:** Inferential statistical indices of the key outcome variables with the factors Time (baseline vs. study end), Group (rTMS vs. sham), and the Time × Group-interaction.

	Inferential Statistics								
	Time			Group			Time × Group Interaction		
	F	$\eta_{p}^{2}$	[ES]	F	$\eta_{p}^{2}$	[ES]	F	$\eta_{p}^{2}$	[ES]
Depression	F(1, 33) = 44.08 ***		0.572 [L]	F(1, 33) = 10.95 ***		0.249 [L]	F(1, 33) = 37.20 ***		0.530 [L]
Fatigue	F(1, 33) = 9.03 **		0.215 [L]	F(1, 33) = 1.75		0.05 [T]	F(1, 33) = 6.80 *		0.171 [L]
Cognitive performance	F(1, 33) = 9.50 **		0.223 [L]	F(1, 33) = 0.19		0.00 [T]	F(1, 33) = 1.28		0.037 [S]
Disability ^1^	F(1, 33) = 10.82 **		0.222 [L]	F(1, 33)= 0.00		0.00 [T]	F(1, 33) = 7.24 **		0.162 [L]

Notes: ^1^ To calculate changes in the EDSS (Expanded Disability Status Scale) scores, an ANOVA was also computed, while scores are reported as medians and ranges. * = *p* < 0.05; ** = *p* < 0.01; *** = *p* < 0.001; [T] = trivial effect size; [S] = small effect size; [L] = medium effect size. [ES] = effect size.

## Data Availability

Upon request, data are made available to experts in the field when reporting clear and reasonable hypotheses, and when credibily ensuring that data are securely stored on a server not otherwise accessible to other persons, and when credibly ensuring that data are not further shared.

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
