# Peer review of "Effectiveness of Repetitive Transcranial Magnetic Stimulation (rTMS) Add-On Therapy to a Standard Treatment in Individuals with Multiple Sclerosis and Concomitant Symptoms of Depression—Results from a Randomized Clinical Trial and Pilot Study"

_jcm, 2023, doi:10.3390/jcm12072525_

Round 1

Reviewer 1 Report

Authors state that this is a RCT, when this is at best low-quality pilot RCT

Many typos and grammatical errors all throughout and the authors used inappropriate terms like “magnetic waves”, “theta stimulation” and “lateral dorsal cortex of the forehead”, which casts a doubt on their experience and expertise. Authors mention a current of “2 mAh”, which is not a concept used in TMS

The randomization procedure is completely outdated and inadequate.

There is no pre-existing literature for the treatment of MDD in MS using rTMS, so it is not possible to make proper sample size calculations for a RCT

No rationale is offered for their protocol. Why only 10 sessions, when usually contemporary trials of TMS use at least 20-30 sessions? It is not even clear what kind of stimulation protocol was used on the left. TBS or non-patterned? If iTBS, why 20 Hz gamma, when usually iTBS is administered at 30 or 50 Hz? How was the targeting done? What about blinding integrity? There is not adequate description of the sham TMS, beyond mentioning it is a “passive device”

Not sure why the authors keep repeating “large effect size” throughout the manuscript, which is inappropriate. The norm is to simply include the statistic itself. There can be an interpretation of the effect size measure, but only in the discussion section.

The limitation section is not serious, statements such as “First, the sample size might be small, though, we mainly focused on effect sizes, which are not sensitive to sample sizes.“ is completely inadequate to justify the methodology used, or lack thereof. Also, nothing stated above is discussed.

Author Response

We thank Reviewer #1 for their candid and valuable comments, which helped us to improve the quality of the manuscript. Please find the detailed point-by-point-response attached as a separate file. 

Thank you once again for your efforts.

Reviewer 2 Report

This study aimed to examine whether rTMS can improve symptoms of fatigue and depression as well as cognitive performance, and decrease the degree of disability in MS in comparison with sham stimulation. This is an interesting study, but the readability of this manuscript is poor and needs language polishing. The discussion is poor.

1.     In line 36, please add the full name of SSRI (Selective Serotonin                       Reuptake Inhibitor).

2.     What mean of rTMS included current 2 mA in line 36? Did you use an             intervention of combined rTMS with tDCS?

3.     In line 74, the manuscript has a punctuation error.

4.     A typo mistake showed in line 81.

5.     Please do not write a sentence as a paragraph from lines 79 to 82 and please give some details.

6.     In line 100, please add the full name of ECT.

7.     The readability is poor from lines 88 to 117.

8.     Please delete the title of the subsection in line 118.

9.     The colon followed by a semicolon in lines 136-147.

10.  In line 136, age followed it with a comma.

11.  A typo mistake showed in lines 157 and 184.

12.  In line 218, articles used repeatedly.

13.  The intervention parameters of rTMS described difference between abstract and method section.

14.  Please give the detail of stimulation frequency in theta stimulation. Why used this stimulation protocol (“The stimulation intensity was initially 30%, and each session increased by 10% to 100%”)? Did you reference frame?

15.  The readability is poor from lines 227 to 228.

16.  The figure 1 showed 17 participates were analyzed in control group and 18 participates were analyzed in intervention group, so you cannot write 40 participates in the abstract and in table 1 and 2.

17.  In table 3, you do not have to write [ES].

18.  Please give the results of post hoc tests between groups in results section.

19.  The discussion is poor.

Author Response

We thank Reviewer #2 for their candid and valuable comments, which helped us to improve the quality of the manuscript. Please find the detailed point-by-point-response attached as a separate file. 

Thank you once again for your efforts.

Round 2

Reviewer 1 Report

None

Author Response

Again, thank you so much for the care devoted to the present manuscript.
